# FROM GUANYIN, UFOS TO PARADISE: CAPTURING CULTURAL VARIATION IN DREAM INTERPRETATION

## ABSTRACT

Humans have long sought to uncover the mystery of dreams from divine signs for predicting fortune and future, to psychology framing them as reflections of the subconscious. This curiosity extends to large language models (LLMs), where commercial LLMs e.g., OpenAI and DeepSeek exhibit preliminary dream interpretation abilities. However, open-source research remains limited to monolingual, western-centric datasets, with evaluations largely confined to classification tasks. We address these gaps by introducing a bilingual dataset of 31,877 unique dream–interpretation pairs spanning three cultural contexts: China, the Middle East and the West, in English and Arabic. Analysis shows <18% of dream symbols overlap across cultures. Chinese symbols emphasize scenario-based activities and figures like *Guanyin*, Arabic symbols reference religion and concepts such as *paradise* and *fasting*, while English symbols draw on technology like *UFOs* and fictional creatures. We evaluated 17 models and found that new state-of-the-art models integrating general-purpose and reasoning modes into one model perform best in reasoning mode, whereas earlier models separating chat and reasoning favor chat settings. While language is not a bottleneck for SOTA models, capturing cultural nuances of under-represented regions e.g., the Middle East remains challenging. Further fine-tuning of six LLMs shows that LoRA benefits larger models, while full-parameter is better for smaller ones. Although SFT equips models with cultural knowledge, post-training knowledge is less stable than pretraining, exhibiting sensitivity to training settings. Data and code are available at `http://URL.withheld.for.review`.

## 1 INTRODUCTION

Dreams have long fascinated humans (Harris-McCoy, 2012). A major turning point came with Freud's theory that dreams express repressed desires and relieve internal tension (Freud, 1900). Subsequent studies analyzed dreams from psychological and neurological relevance (Wamsley & Stickgold, 2011; Wamsley, 2014; Zadra & Stickgold, 2021), connection to memory and consciousness (Siclari et al., 2017), to modern analyses of dream reports documenting recalled dream content by individuals (Domhoff & Schneider, 2008; Laureano & Calvo, 2024). Dream analysis based on dream narrative was initially carried out by human experts (Elce et al., 2021), later augmented by automatic methods leveraging NLP tools from psychological and linguistic perspectives, and now increasingly explored with large language models (LLMs) (Niederhoffer et al., 2017; McNamara et al., 2019; Juncker, 2023; Laureano & Calvo, 2024). See more literature review in Appendix A.

While these efforts have advanced dream understanding, little attention has been devoted to *dream interpretation*, which seeks to derive symbolic, cultural, and contextual meaning from dream content. Most publicly available datasets and studies are centered on English and Western cultures and adopt linguistic, emotional, psychological or biological views to analyze dreams. However, they rarely address the symbolic complexity or cultural variability inherent in dream interpretation.

Our case study shows that <18% of dream symbols overlap across Western, Chinese and Middle Eastern cultures (Section 3). Moreover, even when the same symbol appears, its interpretations diverge substantially. In Figure 1, all three cultures associate *Water* with positive meanings, whereas *Fire* varies widely. In Western culture it signals threat, betrayal or temptation; in Chinese tradition it symbolizes wealth and prosperity; and in Middle Eastern views it reflects both danger and wealth.

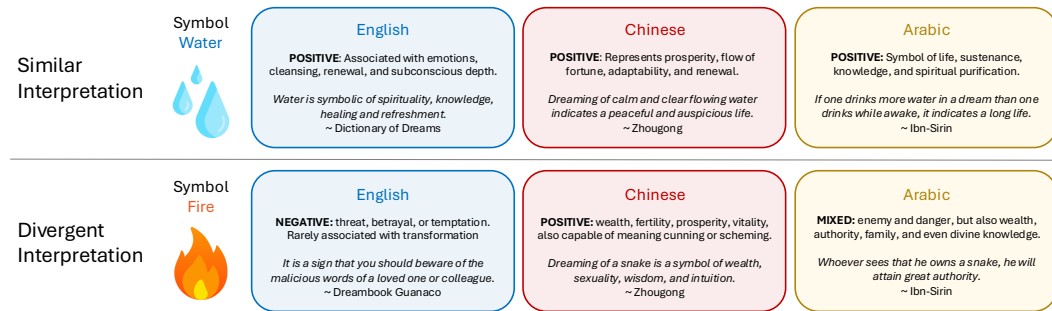

Figure 1: Dream Interpretations of symbols *Water* and *Fire* cross three cultures.

To mitigate this gap, we collect a bilingual dream interpretation dataset from three culture sources — West, China and Middle East, resulting in a total of 31,877 unique entries in Arabic and English languages. Building on these dream-interpretation entries, we formulate four tasks covering free-form question answering (QA), multi-choice question answering (MCQ), is it a *good or bad* dream (GB), and is this interpretation *true or false* (TF). The dataset supports both training and evaluation.

Based on our dataset, we analyzed dream symbols and interpretations across cultures. Evaluations across 17 LLMs show that new state-of-the-art models with two modes (general-purpose and reasoning) in one model outperform earlier models separating these two modes. Current LLMs have overcome language barriers while cultural nuances remain room to improve. Further fine-tuning of six LLMs shows that LoRA benefits larger models, while full-parameter tuning is better for smaller ones. Our contributions are summarized as follows:

- We curated a large-scale bilingual dream interpretation dataset for training and evaluation, covering Western, Chinese and Arabic cultural contexts.
- We analyze cross-cultural similarities and differences in interpreting the same dream symbols, revealing culturally grounded divergence in symbolic meanings.
- We evaluate 17 LLMs and fine-tune six models, revealing the advantage of *one model, two modes*, and the optimal SFT settings for cross-cultural dream interpretation.

## 2 DATASET

Table 1: Dataset statistics of unique pairs and its extended four task formats: QA, MCQ, Good/Bad and True/False across three cultures. Pairs: the cleaned and deduplicated unique symbol–interpretation entries. Train and Test: number of entries.

| Culture | Pairs | QA | MCQ | GB | TF | Train | Test |
|---------|-------|------|------|-------|-------|--------|-------|
| Arabic | 5,568 | 4,466 | 5,568 | 5,568 | 5,568 | 4,466 | 1,102 |
| English | 16,720 | 16,720 | 1,675 | – | – | 15,045 | 1,675 |
| Chinese | 9,589 | 9,589 | 959 | – | – | 8,630 | 959 |

We curated the dataset by collecting raw dream–interpretation pairs from three cultural contexts. For the Middle East, we extracted entries from five sources, primarily books. Western subset was compiled from four sources, with three from Kaggle and one from webpages which required extensive cleaning, preprocessing, and manual verification. Chinese entries were gathered from both webpages and a SQL database. The raw data distribution across sources is shown in Table 6. All entries were then cleaned, deduplicated, consolidated, and manually validated, resulting in 5,568 Arabic, 16,720 English, and 9,589 Chinese unique entries (Table 1). Further details are provided in Appendix B.1.

To prevent data leakage, we split the dataset into training and test at the level of dream symbols rather than dream-interpretation pairs, ensuring that all interpretations of a given symbol remain within the same split. This procedure is applied independently to the Western, Chinese, and Arabic subsets in order to preserve cultural balance.

**Four Task Formulation** From the cleaned dream–interpretation pairs, we categorized dream symbols into 17 groups (Figure 5). For free-form QA and MCQ tasks, we used LLMs to generate culturally specific questions that mimic user queries (Appendix B.2). The gold answer in both tasks

was the original interpretation, while four distractors for MCQ were sampled from interpretations of symbols within the same category as the target symbol. This makes four distractors topically plausible while incorrect, increasing the difficulty of the task and preventing trivial elimination strategies.

In addition to QA and MCQ, users often pose queries such as *I dreamed of a snake yesterday, is it a good sign?* or *I dreamed of a snake, and someone told me it means earning more money, do you think this is true?* To better reflect these real-world inquiry scenarios, we introduce two additional tasks: *(i)* determining whether a dream is a *good or bad* sign, and *(ii)* verifying whether a dream's meaning matches the user's assumption *(true or false)*.

For the Good/Bad task, we used an LLM to label each dream interpretation entry as positive or negative. The True/False task was designed using contrastive reasoning: for each dream symbol, the correct interpretation was paired with a distractor from the same category, yielding QA-style items with one true and one false option. These two tasks were extended only to the Arabic test splits to assess model performance consistency across different inquiry styles.

In each task extension, we emphasized culture-specific interpretive perspectives when designing questions. For example, classical Chinese traditions draw on the five elements (metal, wood, water, fire, and earth: 金木水火土), Yin–Yang (阴阳), and fate (命格). Western questions reflect astrological, zodiac, semiotic, and psychological perspectives. This ensures that the generated questions are aligned with the interpretive context of the source culture while maintaining diversity through multiple distinct questions per entry.

We then used Gemini to translate examples: Arabic to English, Chinese to English and Arabic, and vice versa, ensuring that all cases are available in two languages with identical content. Table 1 summarizes the dataset statistics for each task along with the final training and test splits.

## 3 Dream Interpretations Across Cultures

Each cultural source differs in the number of dream symbols and interpretations it contains. The Arabic source (drawn largely from classical Islamic interpretations) has about 3,390 unique symbols but a total of 5,568 entries, meaning many symbols have multiple interpretations. In contrast, the English source has 8,782 unique symbols (out of 10,601 total entries),[1] and the Chinese source has 8,552 unique symbols (out of 9,589 total entries). The higher ratio of entries to unique symbols in Arabic indicates that the Arabic collection provides multiple interpretations for the same symbol more often (e.g., the symbol *Wife* appears with several distinct interpretations in the Arabic data), whereas the English and Chinese lists typically have fewer entries per symbol. This means the Arabic dream dictionary often offers numerous interpretations for popular symbols, whereas the English and Chinese dictionaries tend to list more symbols but usually with one interpretation each.

### 3.1 Dominant Categories of Symbols in Each Culture

Dream symbols can be grouped into broad thematic categories, such as emotions, animals, family relations, objects, and events/activities. We analyzed the distribution of symbols in each culture across some of these key categories to see which themes are most prominent.

**Animals**  A significant portion of symbols in all three cultures are animals, but this category is especially prominent in the Chinese list. About ∼6% of Chinese symbols are animal-related (e.g. dragons, horses, snakes), compared to ∼3% in Arabic and ∼2% in English. The Arabic compendium does include many animal symbols (e.g. sheep, dogs, birds), often with multiple interpretations, *A Dog*, for example, is interpreted by Ibn Sirin as an enemy or mean person. The Chinese source not only lists the animals themselves but also specific scenarios involving them (like *Hen Laying Eggs* or *Being bitten by a pig*), hence the higher count.

**Emotions**  Explicit emotion symbols (such as *anger*, *love*, *fear*, *crying*) form a relatively small category in all datasets. The English and Arabic lists have around ∼1% of symbols dealing with emotional states, while the Chinese list has about ∼2%. Chinese entries include many nuanced

---

[1]We excluded myislamicdream.com from analysis to ensure accuracy due to its ambiguous origin.

emotional scenarios (for example, *Crying loudly*, *Laughing*, etc.), contributing to a slightly higher share. Overall, though, pure emotion terms are not a large portion of any list.

**Family and Relationships**   Family-related symbols (like mother, father, wife, husband, etc.) appear in all cultures, but they are far more prominent in the Chinese data. About ∼6% of Chinese symbols involve family or close relationships, compared to ∼2% in English and only ∼1% in Arabic. The Chinese source has an abundance of entries about family members and relationship situations – for example, dreaming of a *Girlfriend* or various conditions affecting one's wife or husband are detailed (one entry notes that dreaming of a girlfriend indicates good relationship luck and urges the dreamer to cherish the relationship). The English list also contains family terms (e.g. *Deceased mother-in-law*, *Pregnant wife*), but fewer in number. Arabic interpretations do include core family figures (mother, father, wife, etc.) but relatively sparsely; familial symbols are not as emphasized in the Arabic compilation beyond key roles like parents or spouse.

**Events and Actions**   One of the starkest differences is in the prevalence of dream symbols that are actually actions or events (as opposed to static objects or nouns). The Chinese dream dictionary is rich in scenario-based symbols, roughly ∼32% of Chinese symbols are described as actions or events (often phrased as verb phrases or entire situations). There are countless entries like *Buying [various items]*, *Flying in the sky*, *Carrying a coffin*, *Playing football*, *Arguing*, etc., which indicate that Chinese interpretations often consider the context or activity within the dream as the symbol to interpret. In the English data, only about ∼11% of symbols are action-based (there are entries like *Swimming*, *Running* or *Shooting* but they are fewer). The Arabic source lies in between: around ∼15% of its symbols are events or actions (for example, *Flying* appears as a symbol with many interpretations, and there are entries like *Climbing a mountain* mentioned as related to other symbols). The prevalence of action-oriented symbols in Chinese suggests that Chinese dream interpretation leans heavily toward interpreting entire scenarios and behaviors, e.g. dreaming of studying, fighting, or dancing, whereas Arabic and English sources focus more on static symbols (objects or entities) and interpret the action through context if needed.

**Objects and Other Concepts**   By far the largest category in all three cultures is the broad catch-all of objects, places, and abstract concepts. These make up the remainder (approximately 80–92% of the symbols, when considering symbols not classified as animals, family, emotions, or explicit events). Everyday objects, natural phenomena, body parts, occupations, abstract ideas, and religious concepts fall here. For instance, the Arabic list is dominated by objects and concepts often with spiritual significance or everyday importance: e.g. *Milk*, *Ring*, *Sword*, *Gold*, *Hair* (which alone has 40+ interpretations in Arabic) are common symbols. The English list likewise has many object symbols (from *Tree* and *Car* to *Computer* or *Cake*), and the Chinese list includes not only objects like *Rice*, *Orange*, *Fire* but also very specific cultural objects (e.g. *Jade Bracelet*, *Mahjong tiles*) and concepts. In the Arabic source, a notable subset of these "other" symbols are Islamic religious terms and figures (e.g. *Hajj* (pilgrimage), *Ka'aba*, or names of Quranic chapters like *Al-Bayyina*). In Chinese, a notable subset includes spiritual and folk-religion symbols (Buddhas, Bodhisattvas, temples) as well as everyday life activities. The English data's "other" objects range widely from mundane items (tools, foods, clothes) to modern concepts. For example, *Television* and *Electric Fan* appear in the Chinese and English lists (reflecting modern life influences in dreams), whereas such modern appliances are absent in the classical Arabic source.

## 3.2   Overlap and Uniqueness of Symbols Across Cultures

An important question in cross-cultural dream studies concerns the extent to which different traditions share common symbols. Many motifs, such as the *sun*, *water*, or *dog*, appear across cultures, while others remain distinctive, reflecting particular environments and cultural heritages.

Figure 2 presents the overlaps in unique dream symbols among Arabic, English, and Chinese sources. The diagram shows that **595 symbols** are common to all three cultures (<18%). These shared motifs are typically universal dream images, such as animals (e.g., dogs, cats), natural elements (e.g., fire, rain), and basic human roles (e.g., mother, king). Beyond this shared core, **347 symbols** appear exclusively in Arabic and English, **237** in Arabic and Chinese, and **1,203** in English and Chinese.

Each tradition also maintains a large set of unique symbols: approximately **2,055** are exclusive to Arabic, **6,563** to English, and **6,438** to Chinese. In proportional terms, about 64% of Arabic symbols, 75% of English symbols, and 76% of Chinese symbols do not occur in the other two datasets. This substantial uniqueness underscores the cultural specificity of dream imagery. For example, the Arabic corpus contains numerous Islamic and Middle Eastern references absent in other lists (e.g., *Garments of Jannah*, heavenly attire of Paradise). The English collection reflects Western modernity and psychology with entries such as *Television* and *UFO*. The Chinese dataset includes culturally specific figures and scenarios, such as *Guanyin Bodhisattva* (观音), considered highly auspicious, and everyday activities like *Buying a TV with Girlfriend* or *Playing Mahjong*. These examples demonstrate how cultural context shapes the recording and interpretations.

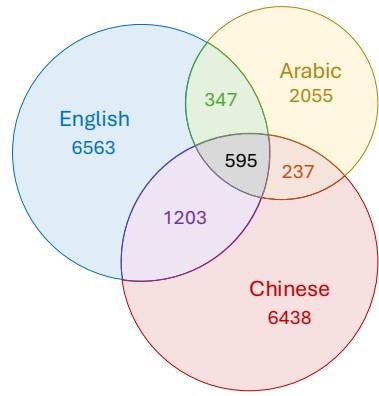

Figure 2: Overlap of dream symbols across Arabic, English, and Chinese cultures.

In sum, while all three traditions share a universal symbolic core, cosmic bodies, animals, fundamental human roles, and natural phenomena, each culture preserves a distinctive repertoire of motifs aligned with its values, practices, and worldview.

**Arabic Culture** Arabic dream interpretations, exemplified by works such as those of Ibn Sīrīn, are deeply rooted in Islamic culture and pre-modern life. Symbols often reference religious practices, historical objects, and concepts such as paradise, fasting, and veiling. Moreover, many entries provide multiple interpretations for the same symbol, reflecting a rich and layered scholarly tradition.

**English Culture** The English corpus, drawing from Western dream dictionaries and modern psychological perspectives, covers a broad range of motifs, including technology, contemporary occupations, and fictional creatures. The emphasis falls largely on objects and straightforward themes, with fewer action-oriented entries. This reflects a secular and modern outlook, where symbols such as *television* or *UFO* receive explicit interpretations.

**Chinese Culture** The Chinese dream symbol tradition is distinguished by detailed scenario-based entries and culturally specific figures. Its extensive catalog includes everyday activities (e.g., shopping, exams, relationship milestones) alongside revered spiritual icons such as *Guanyin* (观音) and historical heroes like *Guan Gong*. Concepts from traditional Chinese philosophy (e.g., *the Five Elements*, *Yin–Yang*) and folk belief also feature prominently. Family relations recur as a central theme, with numerous interpretations involving spouses, in-laws, and pregnancy, highlighting the centrality of kinship in the cultural logic of dreams.

**Summary** Overall, objects and daily life themes dominate all three lists, but animals and family/social elements appear more in Chinese interpretations than in the Arabic or English. Emotional states are occasionally represented explicitly (like *anger* or *joy*) but are relatively infrequent as standalone symbols. Meanwhile, action/event symbols (whole dream scenarios) are a hallmark of the Chinese dataset, whereas the Arabic and English sources concentrate more on static symbols. These patterns suggest that Chinese dream interpretation might emphasize the context and outcome of the dream (what happened in the dream), whereas Arabic and English interpretations focus on the primary subject of the dream (the thing or figure seen).

Each culture's dream symbolism is a reflection of its values, environment, and interpretive traditions, from the spiritual undertones of Arabic dreams, to the individualistic and modern scope of English dreams, to the rich cultural tapestry and pragmatic life scenarios seen in Chinese dreams. Such an analysis highlights how the human experience of dreaming is universal yet colored by cultural lenses, with both overlapping motifs and distinct emphases in each tradition.

## 4 EXPERIMENTS

In this section, we evaluated 17 models with four tasks, including MCQ as the primary, free-form QA, Good/Bad, and True/False. We aim to answer three questions: *(i)* who wins for dream

Table 2: Accuracy (%) of eight LLMs and their reasoning variants on MCQ test sets in two languages. Higher scores between general-purpose and reasoning models are highlighted across three cultures ( Ar–yellow , En–blue , Zh–red , Avg–green ); best per column is **bolded**.

| Model | English MCQ Test | | | | Arabic MCQ Test | | | |
|---|---|---|---|---|---|---|---|---|
| | **Ar** | **En** | **Zh** | **Avg** | **Ar** | **En** | **Zh** | **Avg** |
| Closed-source Models | | | | | | | | |
| GPT5 | 71.7 | 91.3 | **99.8** | 87.7 | 74.4 | 84.5 | 99.6 | 85.4 |
| GPT5-R | 73.9 | **98.0** | **99.8** | 91.3 | 80.7 | 96.8 | 100.0 | 92.9 |
| Claude-Sonnet-4 | 70.9 | 97.7 | 99.6 | 90.3 | 68.9 | 98.4 | 99.7 | 90.0 |
| Claude-Sonnet-4-R | 69.8 | 97.1 | 99.6 | 89.7 | 68.1 | 98.5 | 99.9 | 89.9 |
| DeepSeek-v3.1-Terminus | 61.9 | 97.4 | 99.4 | 87.4 | 65.2 | 95.0 | 98.4 | 87.1 |
| DeepSeek-v3.1-Terminus-R | 68.5 | 96.2 | **99.8** | 88.9 | 71.8 | 93.4 | 99.5 | 88.6 |
| Open-source Models | | | | | | | | |
| Qwen3-8B | 58.0 | 96.1 | 98.8 | 85.6 | 54.4 | 91.6 | 96.6 | 81.9 |
| Qwen3-8B-R | 59.3 | 97.1 | **99.8** | 86.6 | 61.0 | 93.7 | 98.6 | 85.3 |
| Qwen3-1.7B | 51.0 | 87.0 | 89.5 | 77.0 | 46.2 | 74.0 | 71.5 | 65.2 |
| Qwen3-1.7B-R | 50.5 | 92.0 | 95.9 | 80.8 | 47.1 | 80.3 | 81.2 | 70.7 |
| Qwen2.5-7B-Instruct | 47.7 | 95.0 | 98.2 | 81.9 | 48.6 | 89.8 | 94.3 | 78.8 |
| DeepSeek-R1-Distill-Qwen-7B | 43.0 | 84.9 | 91.4 | 74.2 | 33.6 | 40.1 | 37.1 | 37.4 |
| Llama-3.1-8B-Instruct | 40.9 | 95.8 | 97.7 | 80.1 | 26.0 | 68.8 | 78.2 | 58.6 |
| DeepSeek-R1-Distill-Llama-8B | 45.3 | 93.5 | 98.2 | 80.5 | 34.3 | 70.4 | 74.0 | 60.7 |
| Qwen2.5-1.5B-Instruct | 46.2 | 70.0 | 54.4 | 59.0 | 44.9 | 54.9 | 39.3 | 48.0 |
| DeepSeek-R1-Distill-Qwen-1.5B | 33.3 | 53.7 | 58.3 | 48.8 | 23.3 | 23.4 | 22.2 | 23.1 |

interpretation, general large language models (LLMs) versus large reasoning models (LRMs), can Qwen interpret dreams better from the perspective of Zhougong? *(ii)* Can continuous supervised fine-tuning (SFT) with data varying from cultures and languages strengthen models' understanding of dreams? Compared with full-parameter SFT, when would LoRA SFT outperform? *(iii)* Who interprets and predicts the future in a more positive manner, humans or AIs?

### 4.1 EXPERIMENTAL SETUP

**Models**  We include three commercial LLMs and its reasoning variants: GPT-5-2025-08-07, Claude-Sonnet-4-20250514, DeepSeek-v3.1-Terminus (chat/reasoner), as well as 11 open-source models from 1B to 8B: Qwen3-8B/1.7B (chat and reasoning), Qwen2.5-Math-7B and Deeepseek-R1-Distill-Qwen-7B, Llama-3.1-8B and Deepseek-R1-Distill-Llama-8B, Qwen2.5-1.5B-Instruct and DeepSeek-R1-Distill-Qwen-1.5B, Llama-3.2-1B.

**Evaluation Tasks and Metrics**  For the MCQ, Good/Bad and True/False tasks, gold labels are available, and we use accuracy as the evaluation metric. For QA, we employed GPT5-mini as a judge (prompts in Figure 17) to assess correctness and sentiment. Correctness was evaluated by comparing model responses against human interpretations on a 1–5 scale. Sentiment was assessed by first identifying responses as positive or negative, then calculating the percentage of positive ones.

**LoRA and Full-parameter SFT Setups**  We used the model-specific chat template to format the question and corresponding interpretations from our QA training set, and then the cross-entropy loss is computed only on the answer. For both full-parameter and LoRA SFT, we train the model for 1 epoch (if not stated otherwise) with a learning rate of 1e-5. A cosine scheduler is applied with the warm-up ratio as 0.2. We tune the gradient accumulation and per-device batch size to ensure larger (7-8B, 64 accumulation steps with 1 sample per device) and smaller (1-2B, 1 accumulation steps with 64 samples per device) models have an equivalent batch size of 64 per step. For LoRA SFT, we add LoRA adapters on {Q,K,V,O} projections with $r$ and $\alpha$ set as 64 and 16 respectively. All models are trained on two A100 GPUs with bf16 data type.

## 4.2 WHO WINS?

**Do LRMs Outperform LLMs for Dream Interpretation?** In new SOTA LLMs — one model, two modes, reasoning is controlled by an argument, e.g., `reasoning_effort = medium/minimal` for GPT5 and `thinking = enabled/disabled` for Claude. As shown in Table 2, these new models including GPT5, Claude, DeepSeek-v3.1 and Qwen3 demonstrate that their reasoning mode is consistently superior to or on par with their chat mode. This unified design allows models to flexibly combine general knowledge with reasoning, rather than treating them as disjoint capabilities.

By contrast, earlier models which separate chat and reasoning variants (e.g., Qwen2.5-1.5B/7B, Llama-3.1-8B). In these cases, the reasoning versions are often inferior or comparable to the chat versions, as the two cannot effectively benefit from one another. These findings suggest that dream interpretation requires both extensive knowledge and moderate reasoning ability, making integrated reasoning–knowledge models particularly well-suited to the task.

**Do Chinese-Centric Qwen Perform Better in Zhougong Dream Interpretation?** The answer is *No*. Chinese-centric models such as Qwen and DeepSeek do not display a significant advantage on the Chinese (Zh) subset. Most models perform well on both Western and Chinese subsets. Instead, performance on the Arabic (Ar) subset is the decisive factor for overall accuracy, with models showing a 10–50% gap compared to English and Chinese cultures, particularly when Arabic culture interpretations are presented in Arabic. For instance, Qwen3-8B achieves 98.8% accuracy on the Zh subset (in English) but only 54.4% on the Ar subset (in Arabic).

We speculate that while state-of-the-art models have improved in low-resource languages, they still lack cultural nuance optimization. This explains why models, including commercial ones, perform well on Western and Chinese subsets (regardless of presentation language) but fail on Arabic cultural assessments, where underrepresented cultural knowledge remains a bottleneck.

**Language Variation Still Impacts Old Models.** On the identical MCQ test set presented in two languages, commercial models and Qwen3-8B demonstrate strong robustness to language variation, performing similarly on English and Arabic. In contrast, smaller and older models show clear disadvantages on the Arabic test set. This suggests that larger SOTA models above 7B have been specifically optimized towards better multilingual communication, handling Arabic better.

New SOTA models that unify knowledge and reasoning organically within a single model substantially outperform earlier models. For example, Qwen3-1.7B achieves performance comparable to Qwen2.5-7B, while Qwen3-7B performs on par with DeepSeek-v3.1. Also, multilingual understanding and generation have improved, but cultural nuance remains insufficient.

Table 3: Accuracy (%) of Qwen3-1.7B/8B and GPT5 on Arabic-culture test sets across three tasks: MCQ, Good/Bad (GB), True/False (TF), with 1,102 examples each, under three model variants: non-reasoning, LoRA-SFT-All and reasoning.

| Model | English Test | | | Arabic Test | | |
|---|---|---|---|---|---|---|
| | MCQ | GB | TF | MCQ | GB | TF |
| Qwen3-1.7B | 51.0 | 67.6 | 52.5 | 46.2 | 66.8 | 48.9 |
| + LoRA-All | 51.2 | 68.8 | 52.8 | 44.1 | 61.0 | 48.2 |
| Qwen3-1.7B-R | 50.5 | 66.5 | 60.0 | 47.1 | 67.0 | 55.9 |
| Qwen3-8B | 58.0 | 67.1 | 52.7 | 54.4 | 68.4 | 50.3 |
| + LoRA-All | 57.2 | 67.2 | 51.4 | 49.8 | 68.8 | 51.4 |
| Qwen3-8B-R | 59.3 | 69.3 | 58.9 | 61.0 | 70.7 | 55.6 |
| GPT5 | 71.7 | 69.6 | 50.4 | 74.4 | 73.9 | 65.4 |
| GPT5-R | 73.9 | 68.8 | 55.9 | 80.7 | 72.8 | 69.3 |

**Can Different Task Formulations Reveal Consistent Results?** *Yes*. Given that most models achieve over 90% accuracy on the En and Zh subsets, we focus on the Arabic subset to assess consistency across task formulations. As shown in Table 3, the GB and TF tasks largely mirror the MCQ trend: models perform better in English than in Arabic, with the exception of GPT-5, which shows robust performance in both languages.

Overall performance of Qwen3-8B and Qwen3-1.7B follow GPT5, and Table 5 for free-form QA exhibits the same pattern. This confirms that model behavior is consistent across tasks. However, even for the "easier" binary classification tasks (GB and TF), accuracies remain below 74%, highlighting limited knowledge of Arabic dream interpretation logic across models.

## 4.3 FULL-PARAMETER VS. LORA SFT

Table 4: Accuracy (%) of five LLMs fine-tuned by full-parameter SFT vs. LoRA SFT in three training dataset settings: English QA, Arabic QA and their mixture. Evaluation on MCQ test sets.

| Test Set Language | | English MCQ | | | | | | Arabic MCQ | | | | | |
|---|---|---|---|---|---|---|---|---|---|---|---|---|---|
| SFT Training Data | | En | | Ar | | All | | En | | Ar | | All | |
| Model | Cul | Full | LoRA | Full | LoRA | Full | LoRA | Full | LoRA | Full | LoRA | Full | LoRA |
| Qwen3-8B | Ar | 49.5 | 57.3 | 65.4 | 57.4 | 60.0 | 57.2 | 5.3 | 53.6 | 0.5 | 53.8 | 34.6 | 49.8 |
| | En | 97.7 | 96.2 | 97.7 | 96.4 | 98.2 | 96.4 | 92.8 | 91.5 | 70.5 | 91.9 | 94.8 | 92.5 |
| | Zh | 98.5 | 98.8 | 98.7 | 98.8 | 99.3 | 99.1 | 95.5 | 96.5 | 92.7 | 96.7 | 96.2 | 96.8 |
| | Avg | 83.7 | 85.4 | 88.4 | 85.5 | 87.2 | 85.5 | 67.6 | 81.6 | 55.5 | 81.9 | 77.4 | 81.0 |
| Qwen3-1.7BQwen3-1.7B | Ar | 46.2 | 51.2 | 50.6 | 51.4 | 39.9 | 51.2 | 45.8 | 45.9 | 48.2 | 45.2 | 41.7 | 44.1 |
| | En | 94.7 | 87.7 | 92.2 | 88.2 | 93.1 | 89.8 | 84.0 | 74.5 | 83.0 | 76.8 | 80.9 | 78.2 |
| | Zh | 95.6 | 89.5 | 93.7 | 90.2 | 95.6 | 90.7 | 83.5 | 70.9 | 85.2 | 72.5 | 82.5 | 73.5 |
| | Avg | 80.6 | 77.4 | 80.3 | 77.9 | 78.0 | 78.7 | 72.6 | 65.2 | 73.3 | 66.4 | 69.7 | 66.9 |
| Qwen2.5-7B-Instruct | Ar | 38.7 | 48.4 | 50.8 | 48.0 | 42.7 | 49.9 | 50.1 | 48.0 | 53.6 | 47.4 | 51.3 | 46.4 |
| | En | 93.1 | 95.7 | 96.0 | 95.5 | 90.3 | 96.7 | 93.9 | 90.9 | 91.6 | 91.5 | 88.4 | 93.3 |
| | Zh | 90.9 | 98.5 | 96.1 | 98.4 | 93.7 | 98.5 | 82.3 | 94.7 | 92.6 | 95.7 | 90.8 | 96.7 |
| | Avg | 76.5 | 82.5 | 82.7 | 82.3 | 77.1 | 83.4 | 78.0 | 79.2 | 80.6 | 79.6 | 78.1 | 80.3 |
| Llama-3.1-8B-Instruct | Ar | 40.6 | 30.9 | 23.3 | 34.9 | 37.3 | 17.6 | 26.0 | 15.4 | 4.3 | 12.5 | 41.6 | 7.5 |
| | En | 90.6 | 95.6 | 89.1 | 96.4 | 79.8 | 94.2 | 46.3 | 58.3 | 59.0 | 66.0 | 54.9 | 51.1 |
| | Zh | 55.6 | 97.5 | 88.4 | 98.0 | 53.8 | 97.8 | 25.6 | 70.7 | 70.7 | 77.1 | 32.4 | 47.8 |
| | Avg | 66.9 | 77.0 | 69.5 | 78.7 | 60.6 | 72.5 | 35.1 | 48.8 | 39.8 | 53.1 | 45.2 | 37.4 |
| Qwen2.5-1.5B-Instruct | Ar | 55.1 | 46.2 | 46.8 | 45.6 | 55.6 | 45.7 | 45.6 | 43.8 | 44.0 | 43.5 | 46.6 | 42.6 |
| | En | 74.8 | 70.8 | 72.6 | 71.4 | 73.9 | 71.7 | 58.4 | 55.6 | 59.7 | 58.3 | 59.4 | 58.4 |
| | Zh | 50.4 | 53.7 | 49.2 | 54.9 | 48.2 | 52.6 | 34.1 | 39.3 | 36.3 | 41.3 | 32.2 | 38.6 |
| | Avg | 62.8 | 59.2 | 59.0 | 59.6 | 62.0 | 59.2 | 48.5 | 48.0 | 49.1 | 49.6 | 48.7 | 48.7 |
| Llama-3.2-1B-Instruct | Ar | 4.9 | 4.9 | 4.9 | 4.9 | 4.8 | 4.9 | 5.7 | 4.9 | 4.1 | 4.9 | 3.4 | 4.9 |
| | En | 19.0 | 19.2 | 19.1 | 19.2 | 19.1 | 18.9 | 19.0 | 18.9 | 18.9 | 18.9 | 19.1 | 18.9 |
| | Zh | 18.9 | 18.8 | 18.9 | 18.9 | 18.9 | 19.1 | 19.1 | 18.9 | 18.9 | 18.9 | 19.1 | 18.9 |
| | Avg | 14.8 | 14.9 | 14.9 | 14.9 | 14.8 | 14.8 | 15.1 | 14.8 | 14.6 | 14.8 | 14.4 | 14.8 |

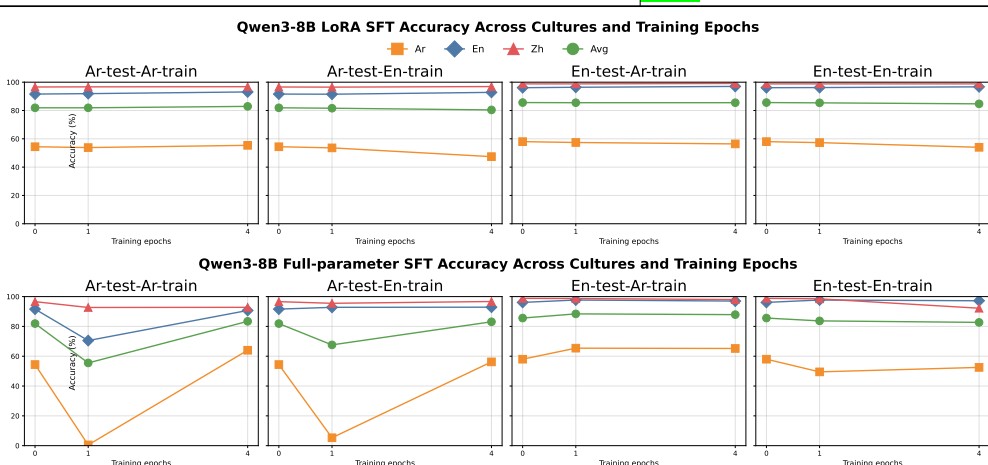

Figure 4: Qwen3-8B Full vs. LoRA SFT accuracy across training epochs and cultures.

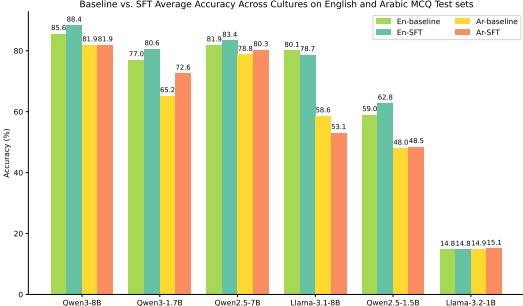

Figure 3: Baseline vs. SFT average accuracy across cultures on English and Arabic MCQ test.

We fine-tuned five open-source models for one epoch under two settings: LoRA and full-parameter using training data in English, Arabic, and a combination of them, each covering all three cultural sources.

**Does SFT Improve Accuracy?** Figure 3 shows that, under appropriate setups, SFT generally improves performance in both languages. As detailed in Table 4, full-parameter SFT is more effective for smaller models (1/1.5B), while LoRA is preferable for larger models (7/8B).

SFT has a limited impact on Western and Chinese subsets due to their already high baselines. The

Arabic subset is the key driver of overall accuracy. Improvements in Arabic culture (yellow cells) largely determine performance gains, where most yellow cells appear when training data is presented in either Arabic language or both (All), indicating that Arabic cultural nuances can be more effectively learned in Arabic language than through English language. Similarly, for Western culture (blue cells), gains are mainly observed with English or bilingual training data.

Table 5: Free-form QA responses' correctness and sentiment under non-reasoning, LoRA-SFT-All and reasoning. QA test set includes 2,634 questions reflecting western and Chinese cultures, presented in two languages. Model responses are more positive then humans' half-half.

| **Model** | **Correct** | | **GT-Sent** | | **AI-Sent** | |
|---|---|---|---|---|---|---|
| | En | Ar | En | Ar | En | Ar |
| Qwen3-1.7B | 3.3 | 2.6 | 50.5 | 52.2 | 61.3 | 56.4 |
| + LoRA-All | 2.7 | 1.9 | 50.5 | 53.0 | 57.3 | 49.6 |
| Qwen3-1.7B-R | 3.4 | 2.8 | 51.6 | 51.4 | 62.7 | 56.8 |
| Qwen3-8B | 3.6 | 3.5 | 50.2 | 52.2 | 60.0 | 56.8 |
| + LoRA-All | 3.2 | 2.7 | 51.1 | 51.1 | 59.4 | 53.7 |
| Qwen3-8B-R | 3.6 | 3.5 | 51.2 | 52.0 | 59.3 | 55.2 |
| GPT5 | 4.1 | 4.1 | 50.5 | 52.7 | 56.7 | 56.5 |
| GPT5-R | 4.1 | 4.2 | 51.7 | 53.2 | 55.4 | 55.7 |
| **Avg** | 3.5 | 3.2 | 50.9 | 52.2 | 59.0 | 55.1 |

However, SFT can also degrade accuracy, particularly when applying full-parameter SFT to larger models (e.g., Llama-3.1-8B and Qwen3-8B), which requires more training epochs to recover disrupted knowledge as below.

**Impact of Training Epochs** Based on Qwen3-8B, we compare the baseline without SFT against full-parameter and LoRA SFT with 1 and 4 training epochs, analyzing the impact of training epochs on dream interpretation across cultures (Figure 4). We find two main results. First, with LoRA SFT, both the number of epochs and the training language have a negligible impact on accuracy when evaluated on English and Arabic MCQ test sets. Second, full-parameter SFT with either English or Arabic data disrupts the model's internal knowledge of Arabic dream interpretation, causing a sharp drop after one epoch, while leaving Western and Chinese subsets largely unaffected. Additional epochs can recover and enhance performance on the Arabic culture subset.

These results suggest that Western and Chinese interpretive traditions are already deeply internalized into parameters during model pretraining, whereas Arabic nuances might be acquired during post-training, which were encoded more shallowly, yielding lower baselines and greater sensitivity to SFT. This aligns with previous findings that knowledge learned during fine-tuning is not well-grounded and tends to induce hallucinations (Gekhman et al., 2024).

### 4.4 ARE AIS MORE POSITIVE THAN HUMANS?

We evaluated three model pairs: Qwen3-7B/1.7B and GPT-5 (chat and reasoning) on 2,634 free-form QA examples spanning Western and Chinese cultures in two languages. As shown in Table 5, human interpretations exhibit no sentiment preference, with positive and negative each holding half. In contrast, model outputs show a slight positive tendency, with 59% positive responses in English and 55% in Arabic.

## 5 CONCLUSION

We introduce a bilingual dream interpretation dataset spanning Western, Chinese, and Arabic cultures, with 31,877 unique entries. We find that <18% of dream symbols overlap across cultures, highlighting the cultural divergence. Using four task formulations, we evaluated 17 LLMs and fine-tuned six models, demonstrating the advantage of the *one model, two modes* paradigm and the capacity of LLMs to handle low-resource languages. However, while models can process the language, they fail to capture cultural nuances, performing markedly worse on Arabic culture interpretations than on Western and Chinese ones. We further identify optimal SFT configurations for models 1-8B, and observe that humans interpret dreams with a roughly neutral balance of positive and negative meanings, whereas models exhibit a slight positive bias. In future work, we plan to focus on enhancing models' sensitivity to cultural nuances and fine-tuning them with reasoning trajectories alongside direct interpretations.

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

## A    BACKGROUND

**Tracing Dream Study**    Dreams have been a topic of human curiosity for centuries, dating back to ancient times. Historical approaches to dream interpretation began with Artemidorus in the 2nd century AD, who systematically studied dream content and proposed interpretive techniques in his work *Oneirocritica* (Harris-McCoy, 2012). A major shift occurred in the 19th century with Sigmund Freud's *The Interpretation of Dreams* (Freud, 1900). He studied that dreams express repressed desires and serve to relieve internal tension, thereby supporting sleep and overall well-being. Later theories

proposed that dreams contribute to emotional regulation and conflict resolution (Cartwright, 2011; Walker & van Der Helm, 2009), memory consolidation (Diekelmann & Born, 2010), and forgetting irrelevant information to enhance learning (Crick & Mitchison, 1983). Some perspectives have also compared dreams to simulations that help individuals prepare for future challenges and threats (Thill & Svensson, 2011).

In recent decades, scientific interest in dreams has increased, particularly in understanding their psychological and neurological relevance. Research has highlighted connections between dreaming and psychophysical health, and pointed to potential roles in sleep-dependent memory processing (Wamsley & Stickgold, 2011; Wamsley, 2014; Zadra & Stickgold, 2021). Furthermore, as internally generated conscious experiences, dreams provide a valuable model for studying the nature of consciousness itself (Nir & Tononi, 2010; Siclari et al., 2017). Despite the growing interest, the processes underlying dream generation and the exact functions of dreams remain partially understood. A major challenge in dream research is the difficulty in quantitatively assessing dream content in a reproducible manner (Elce et al., 2021).

Dream reports, which document the content of dreams recalled by individuals, are essential to the study of dreaming. These reports offer insights into the connection between dreams and waking life (Blagrove et al., 2004) and have long been used as a medium to examine conscious experience during sleep (Nir & Tononi, 2010; Siclari et al., 2017). Thus, many efforts have focused on collecting and analyzing dream narratives or reports (Hall & Van de Castle, 1966; Hauri, 1975; Schredl, 2010).

**Dream Interpretation Datasets**    DreamBank is among the most widely used resources, comprising thousands of annotated dream narratives used in psychological and linguistic research (Domhoff & Schneider, 2008). In addition to public resources, Laureano & Calvo (2024) compiled a proprietary dataset of dreams from 20 patients, with each analyzed by multiple human psychoanalysts and GPT-4. However, manual annotation is time-intensive and typically requires trained human experts, which can hinder the reproducibility and scalability of dream research (Elce et al., 2021).

**Automatic Dream Analysis**    To address these limitations, researchers have increasingly explored the use of natural language processing (NLP) tools to automate the analysis of dream reports. They applied linguistic and computational techniques to analyze dream content. Niederhoffer et al. (2017) examined dream narratives using the Hall and Van de Castle (HVDC) framework, highlighting a higher prevalence of negative emotions, especially sadness in dreams. McNamara et al. (2019) introduced the Dream Content Analysis System (DCAS), which used AI to identify gender-related patterns in dream themes and their relation to mood. Elce et al. (2021) demonstrated that methods such as graph analysis, distributional semantics, and dictionary-based techniques can capture both semantic and structural properties of dream narratives.

Overall, these approaches often rely on models trained on general-purpose corpora such as Wikipedia (Nadeau et al., 2006; Razavi et al., 2014; Altszyler et al., 2017; Sanz et al., 2018; McNamara et al., 2019; Bertolini et al., 2023). However, there are debates regarding how closely dream reports resemble other general types of textual data (Kahan & LaBerge, 2011; Domhoff, 2017; Zheng & Schweickert, 2023). Some evidence suggests that the semantic characteristics of dream reports may diverge significantly from those found in waking narratives (Altszyler et al., 2017). If dream reports are indeed unique in structure and content, the effectiveness of using general-domain NLP models, especially in unsupervised settings, could be substantially limited (Bertolini et al., 2023).

**LLM Dream Analysis**    With the advancement of LLMs, there is growing interest in their use for automated dream analysis. Building on DreamBank, the DReAMy toolkit (Bertolini et al., 2024) offers an open-source framework that leverages multilingual LLMs to automatically annotate dream reports for emotions and characters based on HVDC framework. Blyler & Seligman (2024) used GPT-4 to generate personal narratives and streams of consciousness, while Juncker (2023) employed the model for psychological reflection and dream interpretation. Laureano & Calvo (2024) compared GPT-4's interpretations with those of human analysts, finding that while both were coherent, the AI displayed distinctive linguistic patterns. GPT-4 tended to use more semantic categories such as vision and health-related terms and fewer grammatical elements like impersonal pronouns. Using LIWC features and Naïve Bayes classification, they achieved 99% accuracy in differentiating between the two sources, indicating the significant differences between human-written and LLM-generated dream

analysis. These approaches reduce reliance on manual dream annotation while also expose the gap between human and LLM in dream interpretation.

**Summary**   While these efforts have advanced dream understanding, little attention has been devoted to *dream interpretation*, which seeks to derive symbolic, cultural, and contextual meaning from dream content.

Most publicly available datasets and studies on dream interpretation are centered on English and adopt linguistic, emotional, psychological, or biological perspectives to analyze dream narratives. Such approaches primarily depend on an individual's current condition and recent experiences. By contrast, in cultures such as China, dream interpretation is not only dependent on individuals, but also grounded in centuries of accumulated wisdom and collective observation. Ancient dream-interpretaion records like ZhougongJieMeng provide population-level generalizations, functioning as a form of prior knowledge or statistical distribution of interpretations across thousands of years. These traditions draw on systematic principles such as the Five Elements (metal, wood, water, fire, earth) and Yin–Yang to derive meaning, offering a perspective that extends beyond the individual to collective cultural experience.

To mitigate this gap, we collect a dream interpretation datasets from three culture sources including Arabic, Chinese and Western context, presented in two langauges (Arabic and English). Meanwhile, we examine current state-of-the-art LLMs in dream interpretations across cultures and improved it by continuous training.

# B   DATASET

In this section, we elaborate how we curate the dataset from collecting raw dream-interpretation pairs from three cultural sources, to converting them into four task formulations, to splitting them into training and evaluation benchmark.

## B.1   RAW DATA SOURCES

To construct our dataset, we first collected raw dream–interpretation pairs from a diverse set of cultural traditions and online repositories. The sources span three major cultural contexts: *Arabic*, *English*, and *Chinese*. Within each culture, we curated data from multiple publicly available websites and pre-compiled datasets, ensuring broad coverage of interpretations across different schools and perspectives. Table 6 summarizes the raw data sources and their respective sizes. These raw collections serve as the foundation for subsequent task-specific transformations and benchmark construction.

### B.1.1   ARABIC CULTURE CORPORA

Our dataset combines classical and modern sources of Islamic dream interpretation. Most of the content comes from traditional texts written by well-known scholars in books including **Al–Nabulsi**, **Al–Ihsaei**, **Al–Anbari**, **Ibn Sirin**, and **Ibn Shahin**. We cleaned the data and extracted dream-interpretation entries. Each entry typically consists of a dream symbol paired with an interpretation. When a dream symbol appeared in multiple sources with different interpretations, we retained all available explanations to capture diverse scholarly perspectives, reflecting the natural evolution of dream interpretation from traditional to modern views. After removing duplicates, the dataset includes a total of **5,568** entities

---

[2] https://tafsiralahlam.net
[3] https://www.alanbary.com
[4] https://www.kaggle.com/datasets/manswad/dictionary-of-dreams
[5] https://www.kaggle.com/datasets/yuvrajsanghai/dream-dictionary
[6] https://huggingface.co/datasets/JosephusCheung/GuanacoDataset
[7] https://www.myislamicdream.com
[8] https://www.yourchineseastrology.com
[9] https://www.zgjmorg.com
[10] https://blog.csdn.net/jianghulangzhongshen/article/details/107043786

Table 6: Raw source and size across three cultures. Size refers to numbers after preprocessing.

| Dataset | Lang | Raw Size | Size |
|---|---|---|---|
| tafsiralahlam.net (Al-Ahsa'i) [2] | AR | 441 | 424 |
| alanbary.com (Dr. Khaled Al-Anbari) [3] | AR | 733 | 731 |
| tafsiralahlam.net (Ibn Shahin) [2] | AR | 1,043 | 1,034 |
| tafsiralahlam.net (Ibn Sirin) [2] | AR | 1,096 | 1,088 |
| tafsiralahlam.net (Nabulsi) [2] | AR | 2,291 | 2,291 |
| **Total Arabic Entries** | | | **5,568** |
| Dictionary of Dreams (kaggle/manswad) [4] | EN | 1,040 | 898 |
| Dream Dictionary (kaggle/yuvrajsanghai) [5] | EN | 2,080 | 2,041 |
| Dreambook Guanaco Format (hf/n3rd0) [6] | EN | 9,497 | 7,662 |
| myislamicdream.com [7] | EN | 96,404 | 6,119 |
| **Total English Entries** | | | **16,728** |
| yourchineseastrology.com [8] | ZH | 77 | 77 |
| zgjmorg.com (Zhougong) [9] | ZH | 9,508 | 7,325 |
| Zhougongjiemeng database [10] | ZH | 9,543 | 2,187 |
| **Total Chinese Entries** | | | **9,589** |
| **Total** | | | **31,877** |

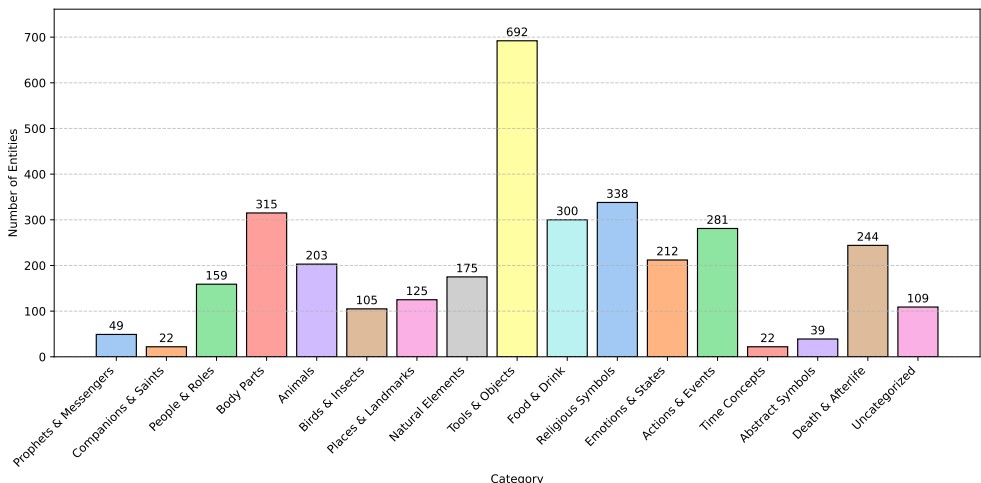

Figure 5: Distribution of Arabic culture dream-interpretation entities across 17 symbolic categories.

**Dream Symbol Category** We applied GPT-4o using prompt shown in Figure 9 to categorize dream symbols into 17 types including prophets and messengers; companions and righteous figures; people and social roles; body parts; animals; birds and insects; places and landmarks; natural elements and phenomena; tools and objects; food and drink; religious symbols and practices; emotional and psychological states; actions and events; time-related concepts; abstract or ambiguous symbols; and symbols related to death and the afterlife. Figure 5 shows the distribution of dream categories, where dream symbols falling into tools and objects are the dominant, followed by religious symbols, food/drinks, action/event and abstract symbols, with occurrence greater than 200.

### B.1.2 ENGLISH CULTURE CORPORA

English culture corpora combine modern, classical, and community-based interpretations. It includes structured resources and digitized texts that cover a range of symbolic meanings and interpretation styles. The raw collection draws from four sources: *(i) Dictionary of Dreams*, 1,041 symbol-

interpretation pairs; *(ii) Dream Dictionary*, 2,080 entries of a similar format; *(iii) DreamBook (Guanaco Format)*, 9,497 user written dreams paired with interpretations generated by a language model; and *(iv)* `myIslamicDream`[2], 96,404 symbolic entries presented in English. The combined raw count is 109,022 entries.

Content from `myislamicdream.com` was collected by a custom Python scraper. The script iterated over paginated tag pages at the pattern `/tags/{page}`, extracted internal links that ended with `.html`, deduplicated links, and then fetched each page. For each page, the parser retained only pages with at least one non-empty paragraph and a minimum visible text length, and it discarded short or empty pages. The crawler used standard request headers and added a one second delay between requests.

**Cleaning and Consolidation**   After collection, we cleaned and consolidated to obtain unique pairs. We first removed empty English interpretations, and then normalized the interpretation text by deleting boilerplate lead-in phrases, such as "Refer to ...", removing parenthetical source notes starting with "(Provided ...)" and deleting footer style material appended at the end of some pages, for instance trailing text "People Who Read This Article Also Read" and embedded domain mentions like "(... www.xyz.org ...)". Afterwards, we excluded entries that were cross references (e.g., "See ...") rather than dream interpretations, and standardized whitespace by removing line breaks and collapsing multiple spaces. Finally, we dropped any residual empty interpretations that arose after text normalization and performed exact match deduplication on the normalized interpretations to consolidate repeated content across sources.

When a symbol appeared in more than one source, we retained distinct interpretations that provided substantive content and removed non-informative duplicates. We collected **16,720** unique dream-interpretation pairs.

### B.1.3   CHINESE CULTURAL CORPORA

The Chinese dataset includes traditional ZhouGong interpretations, structured categorical records, and contemporary astrological perspectives. The collection is drawn from three sources: *(i) Zhougong Dream Dictionary*, parsed from the Zhou Gong website[3], yielding 9,508 entries in Chinese; *(ii) Zhougongjiemeng Database*, reconstructed from a publicly available SQL export[4], containing 9,543 entries in Chinese; and *(iii) Your Chinese Astrology*, an English-language site presenting dream interpretations in a Chinese astrological framework[5], contributing 77 entries.

**Cleaning and Consolidation**   Both the *Zhougong Dream Dictionary* and the *Zhougongjiemeng Database* consist of collections of HTML pages, each processed with a custom parser. During preprocessing, empty or redundant entries were removed, as many of these corresponded to boilerplate pages that merely linked to other content rather than providing substantive interpretations. Because both sources originate from the traditional Zhou Gong corpus, a substantial amount of content was duplicated across them. To consolidate overlapping material, we applied a semantic similarity model (`paraphrase-multilingual-MiniLM-L12-v2`) to compare entries. Pairs with a similarity score above 0.8 were treated as duplicates, and in such cases the longer, more detailed interpretation was retained. Both datasets also underwent additional text cleaning. Redundant lead-in phrases, such as "梦见……意味着什么？and 阅读本文的人还阅读了", were removed using regular expressions to retain only the substantive interpretation content. After this deduplication and cleaning process, the resulting Chinese dataset contained 9,512 unique entries.

The *Your Chinese Astrology* source was processed with a custom parser, which cleaned dream titles by removing boilerplate prefixes (e.g., "Dream Meaning and Interpretation about/of") and extracted interpretation content. The resulting text was normalized and converted into a plain, consistently formatted representation suitable for further processing. This source contributed **77** entries in English. Therefore, there are **9,589** entries in total from Chinese culture.

---

[2]`https://www.myislamicdream.com`

[3]`https://www.zgjm.org`

[4]`https://blog.csdn.net/jianghulangzhongshen/article/details/107043786`

[5]`https://www.yourchineseastrology.com`

## B.2 Four Task Formulation

Based on clean dream-interpretation pairs, we formulated four tasks including free-form question answering (QA), multi-choice question answering (MCQ), is it a *good or bad* dream (GB), and is this interpretation of the dream *true or false* (TF).

### B.2.1 QA Conversion

**Arabic Culture**   To extend the Arabic subset into a question–answer format, we generated culturally grounded questions based on the available symbol–interpretation pairs. For each dream symbol, multiple questions were formulated in five major variants of Arabic languages: Modern Standard Arabic, Gulf, Egyptian, Levantine, and Maghrebi. The prompts were designed to simulate realistic user queries, ranging from conversational expressions to more formal requests, thereby reflecting how speakers across regions might naturally inquire about dream meanings.

Crucially, the answers paired with these questions were always the original interpretations taken directly from the classical Arabic sources. No modifications or paraphrases were introduced: the interpretive content remains identical to that provided by the authoritative texts. The only variation lies in the phrasing of the questions, which differs between dialects.

This procedure produced a dialect-sensitive Arabic QA dataset in which linguistic diversity on the question side is matched with consistent, unaltered interpretations on the answer side. Such a design ensures that the dataset both preserves the symbolic integrity of the source material and provides a realistic testing ground for evaluating models under conditions of dialectal variation in user queries.

**English/Chinese Culture**   To enhance question diversity and cultural relevance, we used LLMs (e.g., `Gemini-1.5-pro`) to generate questions. Given a dream symbol and its associated interpretation, the models were prompted with culturally tailored templates, guiding them to produce natural questions aligned with the interpretive traditions of the source data. See prompts in Figures 7 and 8 in Appendix C.

For the Chinese culture subset, the prompt design emphasizes classical Chinese dream interpretation perspectives, such as 金木水火土, 阴阳 and 命格. English culture templates highlight Western interpretation conventions, interpreting from astrological, zodiac. semiotic and psychological perspectives. Thus, the generated questions are aligned with the interpretive context of the source culture while ensuring diversity (multiple distinct questions per entry). This results in 16,720 QA samples for Western cultural resources and 9,589 QA samples from Chinese culture.

### B.2.2 MCQ Generation

**Arabic Culture**   We extend the Arabic dream interpretation pairs into MCQ format by combining each symbol with its original interpretation and a set of four distractors. The correct answer is always taken directly from the original sources, ensuring that the interpretive content remains unchanged. distractors are sampled from other symbols within the same semantic category, which makes them topically plausible while still incorrect. This category-based sampling increases the difficulty of the task and prevents trivial elimination strategies.

To generate the question text and options, we provide the symbol, its correct interpretation, and the four distractors to an LLM `Gemini-2.0-flash`, together with a structured prompt shown in Figure 10. The model is instructed to produce a single well-formed question in Modern Standard Arabic, accompanied by exactly five answer options (A–E) with the correct answer randomly positioned.

**English/Chinese Culture**   Similar to the procedure of Arabic culture, while we only generated MCQs for the test split of English/Chinese cultures. 10% entries from the Western and Chinese subsets were independently sampled to preserve cultural balance. This results in 1,675 MCQ samples for western culture and 959 for Chinese.

Each sample's correct interpretation is paired with four distractors randomly drawn from other symbols within the same cultural subset. The five options (A–E) are shuffled, with the correct answer randomly positioned to avoid bias, and a post-processing step ensures uniform answer distribution.

Once options are finalized, each entry is passed to `Gemini-1.5-pro` which generates a natural-language question contextualized to the respective cultural framework. The model is supplied with a system prompt that includes both the correct interpretation and its distractors, together with a curated set of culturally appropriate MCQ templates distinct for Western and Chinese dream theories. These templates guide the model to produce diverse and well-phrased questions while maintaining interpretive fidelity. Robustness of the pipeline is ensured via timeout protection, automatic retries, and multithreaded execution.

After expansion through distractor pairing and question generation, the final MCQ dataset contains 2,634 Western samples. All MCQ entries remain traceable to the original dataset through the consistent `id` key. The final dataset includes the dream symbol, interpretation, multiple-choice options, the correct answer label, and the generated question. Detailed prompt structures for Western and Chinese MCQ generation are presented in Figures 11 and 12 respectively in Appendix D.

### B.2.3    GOOD/BAD AND TRUE/FALSE

In addition to free-form questions and straightforward MCQs, users often pose queries such as *I dreamed of a snake yesterday, is it a good sign?* or *I dreamed of a snake, and someone told me it means earning more money, do you think this is true?* To better reflect these real-world inquiry scenarios, we introduce two additional tasks: *(i)* determining whether a dream is a *good or bad* sign, and *(ii)* verifying whether a dream's meaning matches the user's assumption (*true or false*).

For Good/Bad task, we applied Gemini-1.5-Pro to classify each dream-interpretation entry as either good or bad. The input prompt includes the (dream, interpretation) pair, and the model is asked to return the label and brief explanation. This setup encourages the model to reflect on overall sentiment, cultural associations, and emotional tone. For example, a symbol associated with blessings or success would typically be labeled as good, while one linked to fear or misfortune might be labeled as bad.

The True/False task relies on contrastive reasoning. For each dream symbol, we pair its correct interpretation with a distractor interpretation drawn from the same category. Gemini-1.5-Pro is prompted to generate two short QA-style items: one based on the correct interpretation (true), and one based on the distractor (false), written to appear plausible despite being incorrect.

Note that we only extended these two tasks for Arabic culture test splits to evaluate model performance consistency across different inquiry styles. The same approach can be applied to produce more data.

### B.3    TRAINING AND EVALUATION SPLITS

We divide the dataset into training and test splits to enable reliable model development and evaluation. Within each cultural subset, the final processed symbol–interpretation pairs serve as the foundation for multiple task formats.

To prevent data leakage, the splits are created at the level of dream symbols rather than dream-interpretation pairs, ensuring that all interpretations of a given symbol remain within the same split. This procedure is applied independently to the Western, Chinese, and Arabic subsets in order to preserve cultural balance. Table 1 summarizes the dataset statistics for each task and the final splits of the training and test sets.

### B.4    CROSS-CULTURAL DATA ANALYSIS

To better understand the characteristics of our multilingual and multi-cultural dream datasets, we carried out a structured analysis across Arabic, English, and Chinese sources.

**Cross-cultural overlap**    We compared the unique symbols across the three cultures to identify common and distinct concepts. Figure 2 shows that each culture contains a substantial number of culture-specific dream terms, with only a modest overlap across all three. This highlights both the shared foundations and the cultural specificity of dream interpretation traditions.

**Category distributions.**    To provide a high-level view of symbolic content, we grouped dream symbols into broad categories such as *Animals*, *Family & Relationships*, *Emotions*, *Events/Actions*, and *Objects/Other*. Figure 6 illustrates the percentage distribution of these categories per culture. For

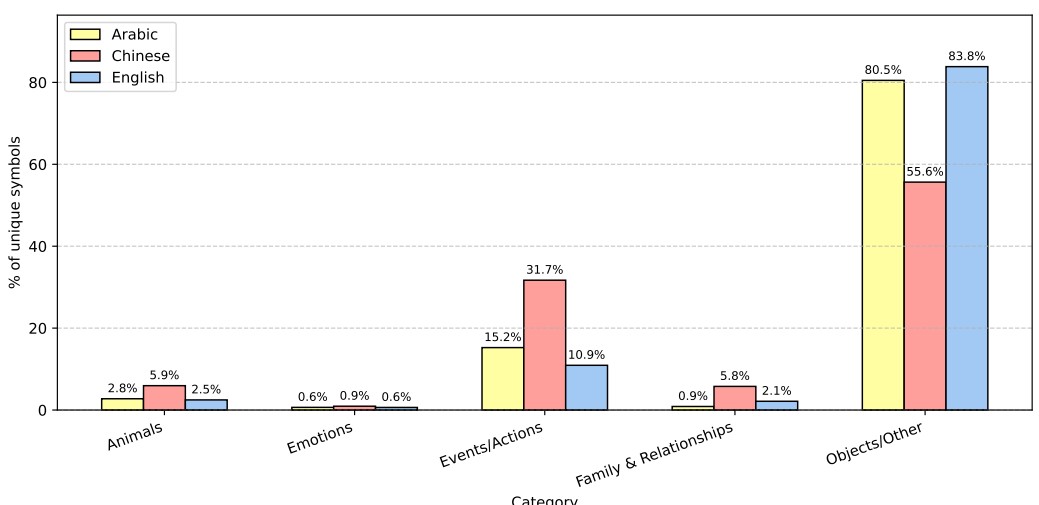

Figure 6: Caption

example, animals dominate a large share of dream entries across all three cultures, whereas emotional and relational terms vary more significantly by language.

## C   PROMPTS FOR QA GENERATION

**You are a helpful assistant to interpret dreams from Western culture and theory.** This is a symbol and interpretation pair in English from Western dream interpretation.
**symbol:** {symbol}
**interpretation:** {interpretation}
Use the following template of questions to generate an appropriate question for each symbol–interpretation pair. You may also generate original questions based on the given content. Start the question directly, without adding introductory text. Both the question and the interpretation should explicitly reference Western culture. Each symbol can result in between one and five questions depending on the interpretation. The output must be returned as a list of questions in JSON format without any additional text or explanation.

- I had a strange dream involving {symbol}. Could it be reflecting something deeper going on in my life?
- When {symbol} appears in a dream, could it point to something unresolved within me?
- I've been thinking about a dream I had with {symbol} in it, what might that say about my current state?
- Could dreaming about {symbol} be linked to recent emotions or stress I've been experiencing?
- What kind of message or warning might a dream about {symbol} be trying to send me?
- If I keep dreaming about {symbol}, could it be a sign that my mind is trying to process something?
- I dreamed about {symbol} last night, could this relate to something I haven't been paying attention to?
- In Western dream interpretation, what might the presence of {symbol} suggest about my inner world?
- What personal insight could I gain from a dream that prominently features {symbol}?
- Could {symbol} in my dream represent something I'm avoiding or afraid to confront?
- What does it typically mean if {symbol} shows up in a dream during a time of change?
- If I see {symbol} in a dream, might it reflect my mindset or relationships lately?
- I'm curious whether dreaming of {symbol} might connect to a personal challenge or transition.
- Is it possible that {symbol} showing up in my dream symbolizes a decision or dilemma I'm facing?
- What could my subconscious be working through if I keep seeing {symbol} in my dreams?
- What does seeing {symbol} in a dream mean?
- What does seeing {symbol} in a dream symbolize in Western culture?

Figure 7: Prompt used with Gemini 1.5 Pro to generate culturally grounded questions based on Western dream interpretations.

**You are a helpful assistant to interpret dreams from Chinese culture and traditions.** This is a symbol and interpretation pair in English from Chinese dream interpretation.
**symbol:** {symbol}
**interpretation:** {interpretation}
Use the following template of questions to generate an appropriate question for each symbol–interpretation pair. You may also generate original questions based on the given content. Each symbol can result in between one and five questions depending on the interpretation. The output must be returned as a list of questions in JSON format without any additional text or explanation.

- What does it mean to dream about {symbol}?
- Can you interpret the meaning of {symbol} appearing in my dream?
- I dreamed about {symbol}, what might this symbolize?
- What is the traditional Chinese interpretation of seeing {symbol} in a dream?
- What could be the deeper meaning of {symbol} in a dream?
- Could dreaming of {symbol} be a sign or omen? What does it represent?
- How should I understand the appearance of {symbol} in my dream?
- What are the possible meanings of dreaming of {symbol} repeatedly?
- I saw {symbol} in my dream and it felt significant, what could it mean?
- What does Chinese dream culture say about dreaming of {symbol}?
- My grandmother used to say dreams carry meanings, what could {symbol} mean if seen in a dream?
- In Chinese folk tradition, how is {symbol} interpreted in dreams?
- How would a traditional dream interpreter explain seeing {symbol}?
- Could the dream of {symbol} indicate something about my future or fortune?
- Is there a symbolic or spiritual meaning to dreaming about {symbol}?
- I'm facing stress lately and dreamed of {symbol}. Could it reflect something in my life?
- I had a peaceful dream with {symbol}, does it reflect emotional or spiritual harmony?
- After dreaming of {symbol}, I've felt uneasy. Could it signal a warning or imbalance?
- What might it mean if I see {symbol} in recurring dreams related to family or work?
- Could the appearance of {symbol} in my dream suggest anything about relationships or health?

Figure 8: Prompt used with Gemini 1.5 Pro to generate culturally grounded questions based on Chinese dream interpretations.

Figure 7 and Figure 8 present the detailed prompt used for curating the dataset for the QA task in Western and Chinese cultures. Gemini-1.5-pro is used to synthesize contextually appropriate questions from the given symbol-interpretation pair and predefined question templates.

## D   PROMPTS FOR MCQ GENERATION

**You are an expert in Islamic dream interpretation.** You will be given a single Arabic dream entity, such as a word or phrase, and your task is to assign it to one of the following high-level categories:
1. Prophets and Messengers
2. Companions, Saints, and Righteous People
3. People and Social Roles
4. Body Parts
5. Animals
6. Birds and Insects
7. Places and Landmarks
8. Natural Elements and Phenomena
9. Tools and Physical Objects
10. Food and Drink
11. Religious Symbols and Practices
12. Emotions and Psychological States
13. Actions and Events
14. Time and Temporal Markers
15. Abstract or Ambiguous Symbols
16. Symbols Related to Death and the Afterlife
17. Uncategorized or Rare Symbols
**Dream Entity:** {entity}
**Output Format (respond with only the category number):**
Category: X

Figure 9: English translation of the prompt used with GPT-4o to categorize Arabic dream entities into one of 17 symbolic categories. The actual prompt was presented in Arabic during inference.

**You are an expert of dream interpretation.** Below is the dream symbol between double ticks "symbol" with its correct interpretation:
''{symbol}'': {interpretation}
And the following are a list of four wrong interpretations of the previous dream symbol:
{wrong_interp}
Using this information, write *one* multiple-choice question about the symbol "{symbol}" using the following rules: • All output should be in Modern Standard Arabic. • Only write one question. • Provide 5 options (A–E), with only one correct answer. • Randomize the position of the correct answer among A–E. • Format your output as valid JSON in the following structure:

```
{
  "question": "....",
  "options": [
    "A) ...",
    "B) ...",
    "C) ...",
    "D) ...",
    "E) ..."
  ],
  "correct_answer": "<correct_answer>" //either "A", "B", "C", "D", or "E"
}
```

Figure 10: Generation prompt for Arabic MCQs. The prompt specifies one correct interpretation and four distractors, and enforces a JSON output format in Modern Standard Arabic.

> **You are a helpful assistant to interpret dreams from Western culture and theory.** This is a set of symbol, interpretation, and options in English from Western dream interpretation.
> **symbol:** {symbol}
> **interpretation:** {interpretation}
> **possible options:** {options}
> The interpretation is the correct answer, while the rest of the options are distractors.
> Use the following template of questions to generate an appropriate multiple-choice question. You can also create an original question based on the symbol, interpretation, and options:
>
> - Which of the following best describes the meaning of dreaming about {symbol} in Western dream interpretation?
> - According to Western dream theory, what does dreaming of {symbol} symbolize?
> - In Western dream interpretation, what might it signify if you dream about {symbol}?
> - Dreaming about {symbol} in Western culture typically indicates which of the following?
> - Based on Western dream interpretation, what is the likely meaning of {symbol} in your dream?
> - If I dream of {symbol}, what would it typically symbolize in Western theory?
> - What could be the deeper meaning of {symbol} in a dream according to Western culture?
> - According to Western dream interpretation, what is the meaning of {symbol} in a dream?

Figure 11: Prompt used with Gemini 1.5 Pro to generate MCQs based on Western dream interpretations.

> **You are a helpful assistant to interpret dreams from Chinese culture and traditions.** This is a set of symbol, interpretation, and options in English from Chinese dream interpretation.
> **symbol:** {symbol}
> **interpretation:** {interpretation}
> **possible options:** {options}
> The interpretation is the correct answer, while the rest of the options are distractors.
> Use the following template of questions to generate an appropriate multiple-choice question. You can also create an original question based on the symbol, interpretation, and options:
>
> - Which of the following best describes the meaning of dreaming about {symbol} in Chinese dream interpretation?
> - According to Chinese dream culture, what does dreaming of {symbol} symbolize?
> - In Chinese dream interpretation, what might it signify if you dream about {symbol}?
> - Dreaming about {symbol} in Chinese culture typically indicates which of the following?
> - Based on Chinese dream interpretation, what is the likely meaning of {symbol} in your dream?

Figure 12: Prompt used with Gemini 1.5 Pro to generate MCQs based on Chinese dream interpretations.

In Figure 9, we present the prompt used for categorizing Arabic dream entities into symbolic categories. Figure 10, Figure 11 and Figure 12 are prompts used for generating the MCQ benchmark dataset in three cultures.

## E  PROMPTS FOR EVALUATION

Figure 13, Figure 14, Figure 15 and Figure 16 present the detailed prompt template on the MCQ, Good/Bad classification, True/False classification and free-form QA tasks, respectively. Figure 17 shows the LLM-as-a-Judge prompt used to evaluate the correctness and sentiment of the model response for the QA task.

**System Prompt**
You are a helpful assistant.

**User Prompt**
**### Question**
{question}

**### Options**
{choices}

=============
Answer the above question by only returning the option (A, B, C, D or E) without any further explanation.

Figure 13: Prompt template for evaluating models on the MCQ task.

**System Prompt**
You are a helpful assistant.

**User Prompt**
If I dream of symbol, is it a good or bad sign? Return "Good" or "Bad" without any further explanation.

Figure 14: Prompt template for evaluating models on the Good/Bad classification task.

**System Prompt**
You are a helpful assistant.

**User Prompt**
**### Question**
{question}

**### Interpretation**
{interpretation}

=============
Given the above question and its interpretation, determine the correctness of the interpretation. Return "True" or "False" without any further explanation.

Figure 15: Prompt template for evaluating models on the True/False classification task.

**System Prompt**
You are a helpful assistant.

**User Prompt**
{question}

Figure 16: Prompt template for evaluating models on the free form QA task.

**System Prompt**
You are a helpful assistant.

**User Prompt**
**### Question**
{question}

**### Ground Truth Interpretation**
{gt_interpretation}

**### Model Interpretation**
{model_interpretation}

=============
Given the above question, ground truth interpretation and model interpretation.
You have three tasks:
1. Determine if the model interpretation is correct (an integer score between 1 to 5, 1 means entirely incorrect, 5 means entirely correct).
2. Determine the sentiment of ground truth interpretation (positive, negative).
3. Determine the sentiment of model interpretation (positive, negative).

Return the results in the following JSON format without any further explanation.
{
"correctness": an integer score between 1 to 5,
"gt_sentiment": "positive" or "negative",
"model_sentiment": "positive" or "negative"
}

Figure 17: Prompt template for evaluating QA task performance with LLM-as-a-Judge approach where GPT-5-mini serves as the judge.

