# OpenReview forum: "From Guanyin, UFOs to Paradise: Capturing Cultural Variation in Dream Interpretation"
_ICLR.cc/2026/Conference — ICLR 2026 Conference Withdrawn Submission_

### Official Review · Reviewer_578P · 2025-10-30

**Soundness:** 1
**Presentation:** 1
**Contribution:** 1
**Rating:** 2
**Confidence:** 4

**Summary:**

The paper curated a dream interpretation dataset for three cultures (Chinese, Arabic and American). It is curated from websites (dictionaries representing scholars e.g., Zhougong, Ibn S¯ır¯ın) and released datasets. Authors constructed a benchmark with four  tasks (good/bad dream, MCQ, True/False, primary) on 17 models (both reaosning and non reasoning models) to see (1) which models have best performances on dream interpretation across cultures (2) the performances between reasoning and non reasoning models (3) does Qwen have better performance on Chinese culture questions. Authors also conducted training experiments (full parameters vs. SFT LoRA) to investigate different hyperparameters relationship with performances of the constructed benchmark

**Strengths:**

S1: Curated and cleaned dataset from online that potentially contains elements to advance cultural studies on dream interpretation. However, the raw sources may involve ethic concern and have been flagged.

S2: Experiments on reasoning and non-reasoning models. Train and compare models with full parameters and LoRA.

**Weaknesses:**

w1: The topic and results gives weak contribution to ICLR community
- the dream interpretation (psychoanalysis) was popular in psychology field during Freud's time. But people were soon losing interests on it since the topic is hard to verify. The topic is very niche and may not be suitable for the ICLR community.
- (table 2) the benchmark curated is not challenging and not suitable to track progress of models capabilities. Most of the models can attain very high scores in the English MCQ and Arabic MCQ already (e.g. the strongest model GPT-5R attains 73.9% for ar, 98% for en, 99.8% for Zh with avg of 91.3% in English MCQ test).
- Recommendation: Submit to other venues for instance specifically for this dream interpretation topic. Adding more dimension analysis -- one possible direction is to use social media data to analyze general community on how they interpret and describe dream. another possible direction is to study the historic records (literatures e.g. poem, records from the scholars about dream). NLP community or cultural analytics community may be interested.

w2: Raw sources are not verified sources (may not be reliable) and they do not have enough representation of overall chosen cultures   (line 140-143)
- Author chose two online website dictionaries claimed to include scholars' interpretations (Ibn S¯ır¯ın for Arabic and Zhougong for Chinese), and one website (Western dream dictionaries for American). However, I double if these websites were reliable and representable for different cultures overall. For instance, the authors' cultural symbols in dreams analysis (e.g. entry / cultural symbol ratio) are based on dictionaries/scholars author chose rather than the 'overall' representation for each culture.
- One way to improve is to find more actual people sharing in social media platform (rather than just the claimed sharings in the dictionaries websites since they could be some designed ads to attract people buying their services). And then cite reliable sources e.g. published book that has collected and translated of scholars' work, religious references/ academic papers (for symbols e.g. Guanyin, Yin–Yang) etc.

w3 unclear methodology and insuiffiecent experiment:
there are many unclear explanation on the methodology (e.g. section 3 classifcation, section 5 finding). Also see more on questions (q1-q2, q4-5).

(i) Classification
- not sure how author classify the different types of cultural symbols. authors did not do thorough works on classifying it. 80-92% of symbols are being classified as "Objects and others", making readers hard to understand the actual differences of symbols between cultures.
- recommend authors using nlp approach on classifying the symbols to help the readers understanding the differences of symbols between different cultures. For instance, using topic model, embeddings to visualize the dataset representation.

(ii) more supports on the finding (one model, two modes paradigm has advantage from line 479)
- the current experiment does not provide sufficient evidence to show it has advantage on the claimed dream interpretation across culture, let alone the general capabilities of models.

w4 Miss relevant works
- some conclusions have been proved and found by other previous works. For instance, the relative poor performance in underrepresented cultures (e.g. Arabic) even for commercial models has been discussed by various works BLEnD (https://arxiv.org/abs/2406.09948), CulturalBench (https://arxiv.org/abs/2410.02677).

w5 poor writing and presentation.
There are many missing details (see questions). Also, the writing contains lengthy and redundant descriptions (e.g. These make up the remainder (approximately 80–92% of the symbols, when considering symbols not classified as animals, family, emotions, or explicit
events).".

**Questions:**

q1. why the GB task only extend to arabic test splits? (line 119-120)

q2. why choose to use Gemini  (line 128)? any test/experiment to justify the choice of gemini?

q3. arabic != islamic. If most of the data is from islamic culture, author should consider to clarify the wording to "islamic culture" directly.

q4. "earlier models which separate chat and reasoning variants (e.g., Qwen2.5-1.5B/7B, Llama-3.1-8B)". Can author explain more on this finding. First, I'm not sure if it is a fair comparison. Authors were comparing the distilled models (deepseek R1 distilled for the two models), which could have other confounding factors (size, model series) rather than just reasoning modes. Second, only Qwen2.5-7B-Instruct shows a (slightly) higher performances than its counterpart in Table 2, which argubly count as significant.

q5. line 467-471: how authors conduct the sentiment analysis.

**Details Of Ethics Concerns:**

Using the raw sources may violate legal compliance. (Appendix B p.13), and the curated dataset does not state which license to release (one raw source (GuanacoDataset requires it to release in the same license GPLv3).

1.  Huggingface (western culture on dream) https://huggingface.co/datasets/JosephusCheung/GuanacoDataset stating "Notice: Effective immediately, the Guanaco and its associated dataset are now licensed under the GPLv3"

2. Chinese websites both https://www.yourchineseastrology.com/ and https://www.zgjmorg.com/ have a bottom row stated Copyright © 2014-2025

From their privacy statement in https://www.zgjmorg.com/site/privacy.html:

The extract:
```
隐私权声明
  来源：周公解梦官网
周公解梦官网（zgjm.org）以此声明对本站用户隐私保护的许诺。

隐私政策
周公解梦官网非常重视对用户隐私权的保护，尊重并保护所有使用服务用户的个人隐私权。为了给您提供更准确、更有个性化的解梦服务，本网站会按照本隐私权政策的规定使用您的相关解梦信息。但本网站将以高度的勤勉、审慎义务对待这些信息。除本隐私权政策另有规定外，在未征得您事先许可的情况下，本网站不会将这些信息对外披露或向第三方提供。本网站会不时更新本隐私权政策。
```

Chatgpt translation:

```
Privacy Statement
Source: Zhougong Dream Interpretation Official Website

The Zhougong Dream Interpretation Official Website (zgjm.org) makes this statement as a promise to protect the privacy of its users.

Privacy Policy
The Zhougong Dream Interpretation Official Website attaches great importance to the protection of users’ privacy rights and respects and safeguards the personal privacy of all users of its services.

In order to provide you with more accurate and personalized dream interpretation services, this website will use your relevant dream interpretation information in accordance with the provisions of this Privacy Policy. However, the website will treat this information with a high degree of diligence and prudence.

Except as otherwise provided in this Privacy Policy, without your prior permission, the website will not disclose or provide this information to third parties.

The website may update this Privacy Policy from time to time.
```

3. Arabic website https://www.alanbary.com/. It states "© 2024 د خالد العنبري"

The extract:
```
© 2024 د خالد العنبري جميع الحقوق محفوظه للمؤلف. ولا يحق اعادة نشر المواضيع او جزء منها الا باذن هذا الموقع لا يستخدم الكوكيز للتبع او لجمع البيانات من المستخدمين
```

Translation (by chatgpt):
```
© 2024 Dr. Khalid Al-Anbari
All rights reserved to the author. It is not permitted to republish the articles or any part of them without permission.
This website does not use cookies to track or collect data from users.
```

---

### Official Review · Reviewer_KoD7 · 2025-10-31

**Soundness:** 3
**Presentation:** 2
**Contribution:** 3
**Rating:** 4
**Confidence:** 4

**Summary:**

The paper introduces a bilingual dataset of 31 877 dream interpretation pairs covering three cultural sources (Western, Chinese, Arabic) and converts it into four task formats (QA, MCQ, Good/Bad, True/False) and analyzes cross-cultural symbol overlap (<18% common symbols) and evaluates different LLMs plus SFT/LoRA variants

The paper finds:
1. One model / two modes models do best.
2. LoRA helps large models while full-parameter helps small ones.
3. Models underperform on Arabic cultural cases.

**Strengths:**

1. To the best of my knowledge I have not come across a similar dataset so I strongly believe in the novelity of the dataset and its scale, assembling 31,877 curated, cross-cultural dream interpretation pairs is a useful resource that fills a gap outside Western-centric datasets.
2. The paper quantitatively measures overlap/uniqueness (e.g., 595 symbols shared by all three cultures <18%) and provides category analyses, which is interesting.
3. The bilingual (English/Arabic) component is a plus, especially since many such resources remain English-centric.

**Weaknesses:**

1. One main concern I had was large parts of the Arabic and Chinese data come from classical/online dream dictionaries (Ibn Sirin, Zhougong, etc.) and scraped webpages; it’s not clear whether authors obtained permission or whether any licensing / copyright issues apply. The paper should state the licenses and whether data release is allowed.

2. The paper repeatedly mentions evaluation of 17 LLMs, but on inspecting the listed models the count appears to be different (commercial models + open source) only. This discrepancy raises questions of clarity and reproducibility: which exact models/variants were used? Are the missing three simply duplicates or omitted? This needs correction.

3. The paper highlights the empirical observation that LoRA fine-tuning benefits larger models while full-parameter fine-tuning benefits smaller ones. While this finding is consistent with the data presented, the claim itself is not novel, it has already been documented in earlier comparative studies such as:
    1. Sun, Xianghui, et al. "A comparative study between full-parameter and lora-based fine-tuning on chinese instruction data for instruction following large language model." arXiv preprint arXiv:2304.08109 (2023).
    2. Shuttleworth, Reece, et al. "Lora vs full fine-tuning: An illusion of equivalence." arXiv preprint arXiv:2410.21228 (2024).

Even though this is not a central claim compared to the dataset contribution, the authors could better position the paper as primarily dataset-centric, emphasizing the cultural and linguistic resource itself while treating the fine-tuning findings as supporting analysis rather than a novelty claim.

4. Evaluation judge for QA: free-form QA correctness and sentiment are judged by GPT5-mini (automated judge). This is convenient but can introduce bias, a small human evaluation subset (with inter-annotator agreement) is required for credibility.

**Questions:**

1. Were all models run with the same prompt templates and temperature settings?
2. Do you plan to host a public leaderboard or benchmark portal (e.g., on Hugging Face or Papers with Code) so future models can be evaluated on this dataset?
3. Beyond confirming previous trends, does your LoRA vs full-parameter analysis reveal any new behavior specific to this domain (e.g., cross-cultural symbolic tasks, low-resource fine-tuning)?
4. Does LLM performance increase with prompt or context optimization stratergies?
4. Can you clarify the licensing status of all the Arabic and Chinese sources?
From my own checks, I could not locate clear open licenses for several of the sources cited in the paper. My current rating is mainly due to remaining questions about the licensing and provenance of the data sources.
That said, I find the dataset itself valuable and timely, and with clearer documentation of licensing as well as broader and more LLM evaluations, I would be more favorable toward an acceptance decision.

**Details Of Ethics Concerns:**

I would like the authors to clarify the copyright and licensing status of the Arabic and Chinese data sources used in the dataset.
Several sources (e.g., tafsiralahlam.net, zgjm.org, and Kaggle dream-dictionary datasets) appear to host or aggregate classical dream-interpretation content, but I could not find clear open licenses or redistribution permissions on those sites.
It would help if the authors could confirm that all data included in the dataset are either publicly licensed for reuse, used under fair academic terms, or covered by appropriate permissions, and to specify under which license the final dataset will be released.

---

### Official Review · Reviewer_BvCz · 2025-10-31

**Soundness:** 3
**Presentation:** 2
**Contribution:** 2
**Rating:** 4
**Confidence:** 3

**Summary:**

This paper introduces a novel, large-scale, bilingual dataset for dream interpretation, covering Western, Chinese, and Arabic cultural contexts. The primary contribution is the dataset itself, which contains over 31,000 dream-interpretation pairs. The authors conduct a cross-cultural analysis, revealing minimal overlap in dream symbols across cultures. They use this dataset to benchmark 17 large language models on several tasks, analyzing model performance, the impact of fine-tuning, and cultural nuances.

**Strengths:**

The paper's strengths lie in its novelty and the considerable effort in curating a multi-cultural dataset for a fascinating domain. The analysis comparing dream symbols and themes across the three cultures provides valuable insights into their differences. The empirical evaluation is reasonably solid, comparing a wide range of modern language models and exploring different fine-tuning strategies (LoRA vs. full-parameter), which yields useful practical findings for the community.

**Weaknesses:**

The most significant weakness is the methodology for dataset creation and evaluation. The data is scraped from public web sources that were likely included in the pre-training corpora of the very models being evaluated. The paper's strategy of splitting the test set by dream symbols does not mitigate the risk of pre-training data contamination. This casts serious doubt on the validity of the zero-shot performance results, as the models may simply be recalling information seen during pre-training rather than demonstrating genuine interpretation or reasoning capabilities. Furthermore, the evaluation lacks engagement with human experts, such as psychoanalysts or cultural anthropologists, relying instead on an LLM for judging the correctness of free-form answers, which is not ideal for such a subjective and culturally deep task.

**Questions:**

Could you elaborate on how you account for the strong possibility of pre-training data contamination, given that the dataset was scraped from public websites? How can we be sure that the zero-shot evaluations are testing reasoning rather than memorization?

Given the deep cultural and psychological complexities of dream interpretation, have you considered validating your dataset and the quality of model-generated interpretations with human experts from the respective cultural and psychoanalytic fields?

---

### Official Review · Reviewer_W6b7 · 2025-11-01

**Soundness:** 2
**Presentation:** 3
**Contribution:** 2
**Rating:** 4
**Confidence:** 2

**Summary:**

Dream-interpretation dataset of 31,877 entries spanning three cultures (Western, Chinese, Middle Eastern) and formulates four tasks: QA, MCQ, Good/Bad, True/False, to study cultural variation in symbolic meanings (e.g., water, fire), finding that <18% of symbols overlap across cultures and interpretations often diverge. they evaluate 17 LLMs, showing that integrated “reasoning mode” variants generally outperform chat-only counterparts, and that Arabic cultural nuance remains hardest despite multilingual robustness.

**Strengths:**

**s1: Originality**
Interesting and creative topic linking cultural psychology and language models; highlights how dream symbolism differs across cultures, which is rarely explored in NLP.

**s2: Dataset design**
Builds a bilingual (English–Arabic) dataset from three cultural traditions (Western, Chinese, Middle Eastern) with four diverse tasks (QA, MCQ, Good/Bad, True/False).

**s3: Comprehensive evaluation** Tests 17 models, comparing reasoning vs. chat modes and different fine-tuning strategies, offering practical insights for future work.

**Weaknesses:**

**w1: Scope** The topic, while unique, is quite niche (dream interpretation), so results may not generalize to broader cultural reasoning tasks.

**Questions:**

**q1: Positioning & background** Please add a concise “Related Work” section in the main paper that contrasts your dataset with the closest NLP/LLM cultural resources and states the differences, you can surface a brief background summary from the appendix into the main paper to anchor readers earlier.

**q2: LLM data validation** Parts of the pipeline use LLMs to generate or label data (e.g., QA/MCQ question generation; Good/Bad classification). Can you report human audit rates or inter-annotator checks to guard against template/judge bias?

**q3: Figure/Table proofreading** Appendix Figure 6 currently has a placeholder caption. Please double-check and correct all figure/table captions before camera-ready.

---

### Note · Authors · 2025-12-28

I have read and agree with the venue's withdrawal policy on behalf of myself and my co-authors.